# Antimicrobial exposure is associated with decreased survival in triple-negative breast cancer

Julia D. Ransohoff [1], Victor Ritter[1], Natasha Purington[1], Karen Andrade[1], Summer Han[1,2,3], Mina Liu[1], Su-Ying Liang[4], Esther M. John[1,3,5], Scarlett L. Gomez[6], Melinda L. Telli [1,3], Lidia Schapira[1,3], Haruka Itakura[1], George W. Sledge[1,3], Ami S. Bhatt [1,7,8] ✉ & Allison W. Kurian [1,3,5,8] ✉

Antimicrobial exposure during curative-intent treatment of triple-negative breast cancer (TNBC) may lead to gut microbiome dysbiosis, decreased circulating and tumor-infiltrating lymphocytes, and inferior outcomes. Here, we investigate the association of antimicrobial exposure and peripheral lymphocyte count during TNBC treatment with survival, using integrated electronic medical record and California Cancer Registry data in the Oncoshare database. Of 772 women with stage I-III TNBC treated with and without standard cytotoxic chemotherapy – prior to the immune checkpoint inhibitor era – most (654, 85%) used antimicrobials. Applying multivariate analyses, we show that each additional total or unique monthly antimicrobial prescription is associated with inferior overall and breast cancer-specific survival. This antimicrobial-mortality association is independent of changes in neutrophil count, is unrelated to disease severity, and is sustained through year three following diagnosis, suggesting antimicrobial exposure negatively impacts TNBC survival. These results may inform mechanistic studies and antimicrobial prescribing decisions in TNBC and other hormone receptor-independent cancers.

Breast cancer is the most common cancer (other than non-melanoma skin cancer) in women globally[1]. The immune system plays an important role in breast cancer outcomes, influencing both overall survival (OS) and breast cancer-specific survival (BCS)[2,3]. Triple-negative breast cancer is the most lethal subtype of breast cancer[4] and disproportionately affects people from racial and ethnic minoritized groups, notably African Americans and Hispanics, and those with lower socioeconomic status[5]. Despite advances in therapies for

TNBC, such as the use of PD-1/PD-L1 immune checkpoint inhibitors (ICI), OS for patients with TNBC is inferior to those with non-triple-negative breast cancer (5 y OS of 64 vs. 81%)[6]. Indeed, of those who are treated pre-operatively with chemotherapy and ICIs, only 60% experience complete response[7], which highlights the importance of identifying biomarkers that predict treatment response and potential interventions that may improve outcomes. Because TNBC lacks hormone receptor and HER2 expression, and thus more closely

[1]Department of Medicine, Stanford University School of Medicine, Stanford, CA, USA. [2]Department of Neurosurgery, Stanford University School of Medicine, Stanford, CA, USA. [3]Stanford Cancer Institute, Stanford University School of Medicine, Stanford, CA, USA. [4]Palo Alto Medical Foundation Research Institute, Sutter Health, Palo Alto, CA, USA. [5]Department of Epidemiology and Population Health, Stanford University School of Medicine, Stanford, CA, USA. [6]Department of Epidemiology and Biostatistics, University of California San Francisco, San Francisco, CA, USA. [7]Department of Genetics, Stanford University School of Medicine, Stanford, CA, USA. [8]These authors contributed equally: Ami S. Bhatt, Allison W. Kurian. ✉e-mail: asbhatt@stanford.edu; akurian@stanford.edu

resembles non-breast cancers, its biology may be most generalizable to other cancer types.

Lower pretreatment absolute lymphocyte count (ALC) is prognostic for breast cancer mortality[8] and early recurrence[9], and lower ALC during breast cancer treatment is associated with inferior disease-free survival (DFS)[10]. In TNBC, which is the most immune-responsive among breast cancer subtypes[2,7,11–17], higher baseline tumor-infiltrating lymphocyte (TIL) density is associated with the achievement of a pathologic complete response (pCR) to pre-operative systemic therapy and improved OS and DFS[2,18]. We previously found that higher ALC is associated with improved survival from TNBC and higher TIL density[19]. Taken together, these findings suggest factors impairing peripheral immunity may adversely impact TNBC treatment outcomes.

There is increasing interest in how the gut microbiota may influence cancer outcomes by modulating host immunity, regulating the tumor microenvironment, and intratumoral microbiome[20]. For example, several studies have demonstrated that the composition of the gut microbiome is associated with ICI response in cancers that are less prevalent than breast cancer, such as locally advanced or metastatic melanoma[21–23]. Additionally, recent small, single-arm phase 1 studies have also demonstrated that fecal microbiota transplantation is sufficient to turn immunotherapy non-responders into responders[24,25].

Antimicrobial[26] and chemotherapy[27,28] exposure disrupt the gut ecosystem, but the impact of these perturbations on breast cancer outcomes is not clear. In a large cohort consisting of >7000 breast cancer cases, antimicrobial exposure within six months preceding diagnosis and treatment with non-ICI-containing regimens was associated with inferior survival, with the strongest signal in the month preceding diagnosis[29]. The impact of antimicrobial exposure during cancer treatment is less clear. On one hand, antibiotic use may eliminate intratumoral microbiota to permit a more immunogenic tumor profile; for example, in a mouse mammary cancer model, tumor infiltration with *Fusobacterium nucleatum* promoted local tumor progression and metastases by suppressing T-cell accumulation in the tumor microenvironment; tumor growth was attenuated with metronidazole[30]. On the other hand, antibiotics have been demonstrated to increase tumor growth in several mouse breast cancer models[31]. There is conflicting evidence as to whether systemic antibiotic use during treatment impairs host immunity, specifically the response to ICI therapy[32]. Several studies have reported an association between antibiotic use and inferior outcomes with ICIs[23,33], independent of antibiotic class used[34], although most of these studies have been performed in patients with melanoma or a variety of non-breast cancers. Germ-free tumor-bearing mice that received fecal microbiome transplants from patients who responded to ICIs demonstrated increased tumor immune cell infiltration[21], suggesting a mechanism by which the microbiome composition influences treatment response. Taken together, it is well known that cancer treatment response may be impacted by the gut microbiome, and animal studies have reported contrasting results on the impact of antibiotics on breast tumor growth. Studies that specifically investigate the role of antibiotics on patient outcomes during the treatment of breast cancer and specifically in TNBC, which is the most immune-responsive subtype of breast cancer, are lacking.

In this work, to investigate whether antimicrobial exposure impacts clinical outcomes in patients with TNBC who are not treated with ICIs (using data that preceded the 2021 change in standard of care for early-stage TNBC to include the use of ICIs), we evaluate the interaction of antimicrobial therapy with time-varying ALC levels and survival over time using a large breast cancer database well-suited for focus on TNBC. The Oncoshare database integrates electronic medical records (EMR) and the California Cancer Registry (CCR, the state's population-based registry) data for breast cancer patients treated in the community and academic healthcare systems since 2000[35,36]. Our hypothesis was that increasing antimicrobial exposure during curative intent TNBC treatment and follow-up adversely impacts survival by impairing lymphocyte-mediated antitumor immunity.

## Results

### Patient characteristics

A total of 799 female patients were diagnosed with stage I-III TNBC from January 2000 to May 2014, treated primarily at Stanford University or Palo Alto Medical Foundation (Sutter Health), and observed for a minimum of 5 years. Patients were excluded if they were immunocompromised before breast cancer diagnosis (n = 21) or had an unknown race and/or ethnicity (n = 6). The final analytic sample consisted of 772 patients (Supplemental Fig. 1). Baseline sociodemographic and clinical characteristics and treatment courses are described for ever- and never-antimicrobial users. Ever antimicrobial users were more likely to have a non-normal body mass index (BMI), be treated in the community versus academic practice, undergo unilateral or bilateral mastectomy versus lumpectomy, have lower minimum ALC or absolute neutrophil count (ANC), and receive growth factor support (Table 1).

The median overall follow-up time (including time to death) was 104 months (interquartile range (IQR) [61.7, 147]); the median follow-up time among those alive through the observation period was 121 months (IQR [87, 161]). Most patients (654, 84%) used antimicrobials after diagnosis. Nearly all exposed patients received antibiotics (649, 99%), and some of these patients received both antibiotics and antifungals (153, 24%); very few received only antifungals (5, 0.8%). There were 24/118 (20%) deaths among patients who never used antimicrobials and 153/654 (23%) deaths in patients who ever used antimicrobials during the study period.

### Inverse probability weighting to estimate probabilities of antimicrobial use

The Cox proportional hazards model for time to any antimicrobial usage yielded adjusted hazard ratios (HRs) with 95% confidence intervals (CIs), and the mixed effect Poisson regression models for the number of total and unique antimicrobials yielded the expected changes in the number of exposures. Higher minimum ALC, evaluated in month-intervals from diagnosis, was associated with a lower likelihood of any antimicrobial exposure; growth factor use and unilateral mastectomy compared to lumpectomy were associated with a higher likelihood of antimicrobial exposure for three definitions of antimicrobial use: any, cumulative total, and cumulative unique antimicrobial exposures (Supplemental Tables 2–4). Evaluation of the mean, standard deviation, minimum, and maximum stabilized weights for each exposure definition model were consistent with no evidence of non-positivity. For inverse probability weighting for any antimicrobial exposure definition, the mean stabilized weight was 1.0 and the standard deviation of the stabilized weights was 0.08; the minimum and maximum weights were 0.7 and 1.5, respectively. For total antimicrobial exposure, the mean was 1.1 and the standard deviation was 0.6; the minimum and maximum were 0.4 and 3.2, respectively. For unique antimicrobial exposure the mean was 1.1 and the standard deviation was 0.3; the minimum and maximum were 0.6 and 3.2, respectively.

### Associations of antimicrobial exposure with survival

The number of total and unique antimicrobial prescriptions ranged from 0 to 59 and 0 to 26, respectively, over the observation period (Supplemental Fig. 2). In a marginal structural Cox regression model (MSM), any antimicrobial use was not associated with BCS (hazard ratio (HR) 1.39, 95% confidence interval (CI) (0.84–2.32)) or OS (1.46 (0.93–2.29)), while cumulative total and unique antimicrobial exposure were associated with inferior BCS (1.05 (1.01–1.08)) and 1.18 (1.13–1.24) for each additional total or unique monthly prescription,

respectively) and OS (1.05 (1.02–1.08) and 1.17 (1.12–1.23) for each additional total or unique monthly prescription, respectively); Fig. 1 and Supplemental Table 5). We summarized the number of patients still at risk of death at each time point by exposure group (Fig. 2).

**Associations of tumor and patient characteristics with survival**
Using marginal structural models (MSMs) to estimate the associations of individual covariates with mortality, we found, for all three exposure definitions, that higher cancer stage and undergoing unilateral mastectomy were associated with decreased OS and BCS (for any antimicrobial exposure: OS for cancer stage III versus I, 4.02 (2.42–6.69); OS for unilateral mastectomy versus lumpectomy, 1.60 (1.10–2.34); BCS for cancer stage III versus I, 6.33 (3.28–12.21); BCS for unilateral mastectomy versus lumpectomy, 1.90 (1.22–2.96); results were similar for total and unique antimicrobial use.

**Sensitivity analysis**
To address whether patients who are sicker at baseline or become frailer during treatment may have inferior outcomes related to their clinical performance status and not due to greater antimicrobial exposure, we considered disease severity, defined by receipt of intravenous antimicrobials and ICD10 codes associated with acute illness (sepsis, severe sepsis, and systemic inflammatory response syndrome) 90 days prior to or following antimicrobial prescriptions (in $n = 54$ patients), as an additional covariate in the MSM model. We did not observe a substantial change in OS or BCS estimates for any exposure definition (for any antimicrobial exposure: OS 1.45 (0.93–2.28), BCS 1.40 (0.84–2.34); for total exposures: OS 1.05 (1.02–1.08), BCS 1.05 (1.02–1.09); for unique exposures: OS 1.16 (1.11–1.21), BCS 1.17 (1.12–1.23)) (Supplemental Table 6).

To evaluate whether the observed mortality impact is ALC-specific, we compared the standard adjusted unweighted Cox regression model to the MSM. When accounting for the effect of ALC in the MSM, we observed a decrease in the HRs for both OS and BCS (OS for unweighted Cox 1.54 (0.98–2.49) versus OS for MSM 1.46 (0.93–2.29); BCS for unweighted Cox 1.47 (0.89–2.45) versus BCS for MSM 1.39 (0.84–2.32)), suggesting that ALC is associated with both outcomes. When accounting similarly for the effect of ANC, we observed no change in the HRs for OS or BCS (OS for unweighted Cox 1.58 (1.01–2.49) versus OS for MSM 1.58 (1.01–2.48); BCS for unweighted Cox 1.53 (0.92–2.54) versus BCS for MSM 1.52 (0.92–2.53)), suggesting ANC is not along the causal pathway from antimicrobial exposure to increased mortality (Supplemental Table 7).

**Associations of antimicrobial exposure with survival over time**
Given that patients with TNBC have an elevated risk of recurrence in the first 2–5 years post-diagnosis, we used landmark analysis to determine whether antimicrobial use was significantly associated with mortality at yearly intervals following diagnosis. This analysis considers the impact of ongoing exposure to antimicrobials after completion of breast cancer treatment on the subset of women who remain event-free but at risk of recurrence at each time point. For those who met the cumulative exposure definitions associated with inferior survival, we found evidence of a strong and sustained by-year association through year 3 post-diagnosis that then decreased at years 4 and 5 (Fig. 3).

**Exploratory analysis of baseline TIL density, antimicrobial exposure, and response to chemotherapy**
We sought to explore whether antimicrobial exposure is associated with TIL density, a known marker for response to chemotherapy. We had access to pretreatment stromal TIL (sTIL) density data for a subset of 53 patients treated with neoadjuvant chemotherapy in a clinical trial[37]. No association was found between baseline sTIL level and subsequent receipt of antimicrobials in this small sample. Increasing

**Table 1 | Baseline sociodemographic and clinical characteristics of triple-negative breast cancer patients by antibiotic use after diagnosis**

| Antibiotic usage | | | | |
|---|---|---|---|---|
| Characteristic | Ever user N = 654 | Never user N = 118 | Overall N = 772 | p value |
| Age at diagnosis, N (%) | | | | 0.77 |
| 21–34 | 48 (7.3) | 11 (9.3) | 59 (7.6) | |
| 35–49 | 226 (35) | 44 (37) | 270 (35) | |
| 50–64 | 243 (37) | 40 (34) | 283 (37) | |
| 65–95 | 137 (21) | 23 (19) | 160 (21) | |
| Socioeconomic status quintile, N (%) | | | | 0.38 |
| 1 | 25 (3.8) | 7 (5.9) | 32 (4.1) | |
| 2 | 64 (9.8) | 10 (8.5) | 74 (9.6) | |
| 3 | 89 (14) | 17 (14) | 106 (14) | |
| 4 | 123 (19) | 29 (25) | 152 (20) | |
| 5 | 353 (54) | 55 (47) | 408 (53) | |
| Race and ethnicity, N (%) | | | | 0.46 |
| Non-Hispanic White | 477 (73) | 78 (66) | 555 (72) | |
| Hispanic | 54 (8.3) | 11 (9.3) | 65 (8.4) | |
| Asian/Pacific Islander | 94 (14) | 23 (19) | 117 (15) | |
| Black | 29 (4.4) | 6 (5.1) | 35 (4.5) | |
| Body mass index*, N (%) | | | | 0.012 |
| Underweight | 12 (1.8) | 0 (0) | 12 (1.6) | |
| Normal | 217 (33) | 39 (33) | 256 (33) | |
| Overweight | 113 (17) | 15 (13) | 128 (17) | |
| Obese | 99 (15) | 9 (7.6) | 108 (14) | |
| Unknown | 213 (33) | 55 (47) | 268 (35) | |
| Ever publicly insured, N (%) | 135 (21) | 28 (24) | 163 (21) | 0.45 |
| Place of care, N (%) | | | | 0.007 |
| Community | 315 (48) | 41 (35) | 356 (46) | |
| University | 339 (52) | 77 (65) | 416 (54) | |
| Stage, N (%) | | | | 0.91 |
| I | 222 (34) | 39 (33) | 261 (34) | |
| II | 325 (50) | 61 (52) | 386 (50) | |
| III | 107 (16) | 18 (15) | 125 (16) | |
| Tumor grade, N (%) | | | | 0.17 |
| 1 | 21 (3.2) | 5 (4.2) | 26 (3.4) | |
| 2 | 122 (19) | 14 (12) | 136 (18) | |
| 3 | 479 (73) | 96 (81) | 575 (74) | |
| Unknown | 32 (4.9) | 3 (2.5) | 35 (4.5) | |
| Surgery type, N (%) | | | | 0.013 |
| Lumpectomy | 323 (49) | 77 (65) | 400 (52) | |
| Bilateral mastectomy | 93 (14) | 14 (12) | 107 (14) | |
| Unilateral mastectomy | 221 (34) | 26 (22) | 247 (32) | |
| No surgery | 17 (2.6) | 1 (0.8) | 18 (2.3) | |
| Received chemotherapy, N (%) | 522 (80) | 87 (74) | 609 (79) | 0.14 |
| Received radiotherapy, N (%) | 359 (55) | 70 (59) | 429 (56) | 0.37 |
| Germline BRCA1/2 pathogenic variant status, N (%) | | | | 0.96 |
| Negative/VUS | 140 (21) | 24 (20) | 164 (21) | |
| Positive | 51 (7.8) | 9 (7.6) | 60 (7.8) | |
| Untested | 463 (71) | 85 (72) | 548 (71) | |

**Table 1 (continued) | Baseline sociodemographic and clinical characteristics of triple-negative breast cancer patients by antibiotic use after diagnosis**

| Antibiotic usage | | | | |
|---|---|---|---|---|
| **Characteristic** | **Ever user N = 654** | **Never user N = 118** | **Overall N = 772** | **p value** |
| Ever low absolute neutrophil count (<1 K/uL) post-diagnosis, N (%) | 142 (22) | 10 (8.5) | 152 (20) | <0.001 |
| Minimum absolute neutrophil count (K/uL) post-diagnosis, Median (IQR) | 2.12 [1.12, 3.14] | 2.70 [1.83, 3.79] | 2.26 [1.24, 3.30] | <0.001 |
| Ever low absolute lymphocyte count (<1 K/uL) post-diagnosis, N (%) | 439 (67) | 48 (41) | 487 (63) | <0.001 |
| Minimum absolute lymphocyte count (K/uL) post-diagnosis, Median (IQR) | 0.70 [0.40, 1.10] | 1.20 [0.70, 1.67] | 0.76 [0.46, 1.20] | <0.001 |
| Ever used growth factor support, N (%) | 287 (44) | 27 (23) | 314 (41) | <0.001 |
| Number of growth factor uses, Median (IQR) | 3.00 [2.00, 4.00] | 2.00 [2.00, 3.00] | 3.00 [2.00, 4.00] | 0.004 |
| Follow-up time (months), Median [IQR] | 104 [62, 146] | 111 [58, 159] | 104 [62, 147] | 0.76 |

*VUS* variant of unknown significance, *IQR* interquartile range.

*Body mass index was calculated as kg/m², where kg is weight in kilograms, and m² is height in meters squared. Underweight was categorized as BMI under 18.5 kg/m²; normal as BMI greater than or equal to 18.5 to 24.9 kg/m²; overweight as BMI greater than or equal to 25 to 29.9 kg/m²; obese as BMI greater than or equal to 30 kg/m².

Two-sided unadjusted p values for comparisons between ever and never users were derived from Pearson's Chi-squared test, Fisher's exact test, or Wilcoxon rank-sum test.

continuous sTIL score was associated with the achievement of a pathologic complete response (pCR) to chemotherapy (median sTIL score 3.00 (IQR 1.75-4.25) for pCR versus 1.00 (0.50–2.00) for non-PCR, *p* = 0.027). In an exploratory analysis of 28 of these patients who had documented pathology outcomes after neoadjuvant chemotherapy, antimicrobial exposure during treatment was not associated with pCR status.

## Discussion

We studied long-term OS and BCS associated with antimicrobial exposure in 772 TNBC patients treated with curative intent from 2000–2014 at two institutions representing community and academic practice. Consistent with the study's hypothesis, we found that cumulative antimicrobial exposure after diagnosis of early-stage TNBC was associated with inferior OS and BCS. This association was related to time-varying ALC levels, was sustained in strength through year 3 post-diagnosis, and was independent of severe illness. To our knowledge, this is the first study to report an association of antimicrobial exposure during treatment with breast cancer outcomes and to explore the interaction of antimicrobials with ALC, which is prognostic in TNBC[19]. If confirmed, these findings may inform antimicrobial prescribing practices during TNBC treatment and follow-up.

We previously reported that lower ALC after TNBC diagnosis was associated with inferior OS and BCS, that the strength of this association increased with time, and that higher baseline TIL density was associated with higher peripheral lymphocyte count during treatment[19], suggesting peripheral lymphocyte count may be a biomarker of tumor-directed immunity. Higher TIL density in pretreatment breast cancer biopsies is also associated with a better response to neoadjuvant chemotherapy[2] and improved DFS and OS in patients

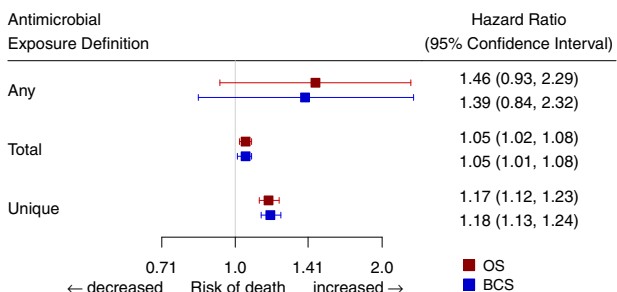

**Fig. 1 | Forest plot of multivariable-adjusted hazard ratios (HRs) and 95% confidence intervals for any, total, and unique antimicrobial exposures in the marginal structural Cox proportional hazards multivariate model (MSM) for N = 772 independent patients.** Data are presented as the HRs ± 95% confidence intervals, reflecting the risk of death for any antimicrobial use, defined as ever versus never receipt of antimicrobials, and the risk of death for each change in the cumulative number of per-month prescriptions for total and unique exposures during observation. MSMs were adjusted for the following variables: age at diagnosis, race, ethnicity, socioeconomic status quintile, cancer stage, tumor grade, receipt of chemotherapy, receipt of radiotherapy, ever use of growth factor support, and surgery type.

with node-positive TNBC[38], which has the most aggressive biology among breast cancer subtypes in terms of early recurrence and worst prognosis[4], and disproportionately affects women of African American and Hispanic race and ethnicity and of lower socioeconomic status[5]. A recent epidemiological study showed impaired survival in breast cancer patients receiving antibiotics prior to diagnosis and treatment without ICIs, and most significantly in the month preceding diagnosis[29], which lends support to the hypothesis that antimicrobial exposure may impair baseline tumor-directed immunity and perhaps TIL levels. Such at-risk patients, however, can only be identified retrospectively. In an exploratory, limited subset analysis, we confirmed that TIL levels at diagnosis were associated with a pathologic complete response to chemotherapy, as others have shown[2], but were not associated with subsequent receipt of antimicrobials and antimicrobial exposure was not associated with pathologic complete response, though the current study's sample size was limited due to a small number of historical samples available for TIL analysis and not sufficiently powered to detect these associations. Taken together, these findings suggest that antimicrobial exposure both before and during treatment is an immune-modulating risk factor for TNBC mortality, but that the relationship between antimicrobials, circulating and tumor-directed lymphocytes, and response to treatment is complex and dynamic. The impact of ICIs on these relationships will be an intriguing area of future study.

While lymphopenia may initially occur during receipt of cytotoxic chemotherapy and predispose patients to opportunistic infections[39] and inferior outcomes[19], emerging data suggest antimicrobial exposure may sustain impaired peripheral immunity secondary to gut microbiota disruption. This altered immunity may contribute to both cancer development[40] and progression[41]. An active area of focus is the augmentation of host antitumor immunity: immunotherapy increases the rate of pathologic complete response, which is correlated with improved survival, when added to neoadjuvant chemotherapy[7] and is also effective in heavily pre-treated TNBC[13]. Breast cancer intratumoral vaccination strategies are also being studied (ClinicalTrials.gov[42] identifiers: NCT02018458, NCT01703754, and NCT02423902). The hypothesis that gut microbiome dysbiosis impairs the efficacy of ICIs is increasingly relevant for early-stage TNBC patients, given the 2021 United States Food and Drug Administration approval of the ICI pembrolizumab with neoadjuvant chemotherapy based on the KEYNOTE-522 trial[7]. The present findings demonstrate an association of increasing antimicrobial exposure with inferior TNBC outcomes

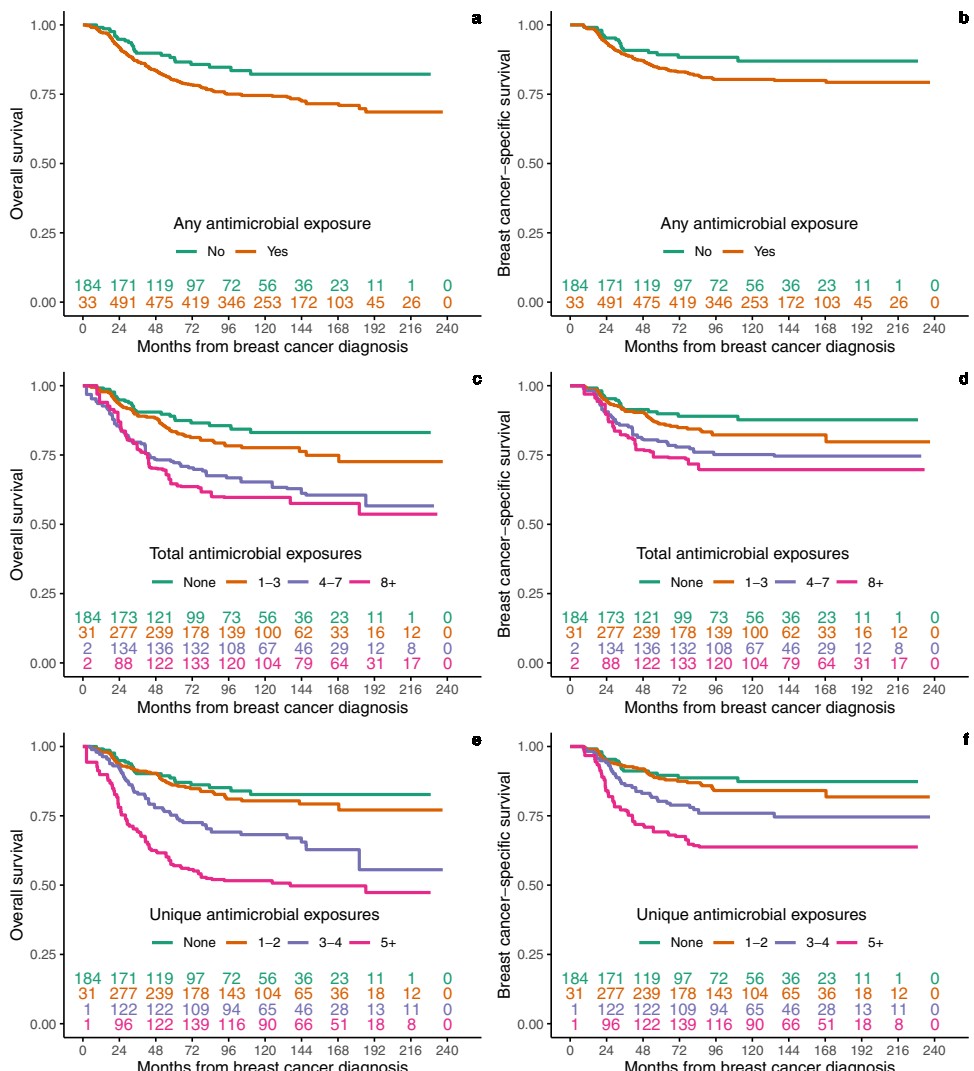

**Fig. 2 | Weighted Kaplan–Meier estimates of overall and breast cancer-specific survival by any, total, and unique number of antimicrobial exposures for N = 772 independent patients. a, b** Overall survival (OS) and breast cancer-specific survival (BCS) estimates for any antimicrobial exposure. **c, d** OS and BCS estimates for total antimicrobial exposure, visualized into quartiles. **e, f** OS and BCS estimates for unique antimicrobial exposure, visualized into quartiles. Inset numbers represent the number of patients still at risk of death at each time point. Patients could move to increasing exposure categories over time.

related to lymphopenia in a population treated before ICI approval in breast cancer and suggest that antimicrobial exposure also impairs the efficacy of traditional cytotoxic chemotherapies. They also offer evidence that lymphopenia may be secondary to antimicrobial-mediated gut microbiome dysbiosis.

This study has limitations that warrant consideration. Since this is a retrospective, observational study using CCR data linked to EMR data collected during the routine course of care, we cannot infer causality and could not collect corollary microbiome samples from the analytic sample of nearly 800 TNBC patients treated since 2000. TIL scores at diagnosis were available for a small subset of patients enrolled in a clinical trial[37], but were not available after chemotherapy and antimicrobial exposure for most patients. Some data are missing, including prescriptions and laboratory results obtained outside of the studied healthcare systems. Because prophylactic antimicrobials are rarely given in general medical practice and are not given in breast cancer, treatment was assumed to be for clinical infection, and we assumed that patients took medications as prescribed. We could not evaluate non-adherence and did not evaluate ICD codes associated with infection proximal to antimicrobial prescriptions, as prior studies have shown limited accuracy in clinical recording, especially of

secondary diagnoses (e.g., infection as a secondary condition diagnosed in a physician office visit focused on breast cancer treatment)[43]. Most infections in adults are treated empirically (i.e., without culture-based evidence of infection with a specific organism) and are cured in the outpatient setting. Thus, we did not evaluate for chart documentation of infection with specific organisms nor development of antimicrobial resistance, as <1% of our sample was treated in the inpatient setting with intravenous antimicrobials where organisms and their antibiotic sensitivities are typically identified. We have also made several assumptions in statistical modeling. To address the consistency assumption, we evaluated three well-defined antimicrobial exposure definitions and separately evaluated mortality associations with each. We assumed that exposed and unexposed individuals had equivalent distributions of mortality predictors but acknowledge that there remain unmeasured confounders that may impact exposure and mortality associations, for which we cannot formally test given the observational nature of the study.

Although it is possible that excluded patients without recorded blood counts were healthier, the finding of a survival association with ALC but not ANC suggests a lymphocyte-specific observation, even though patients with both low ALC and ANC values were more likely

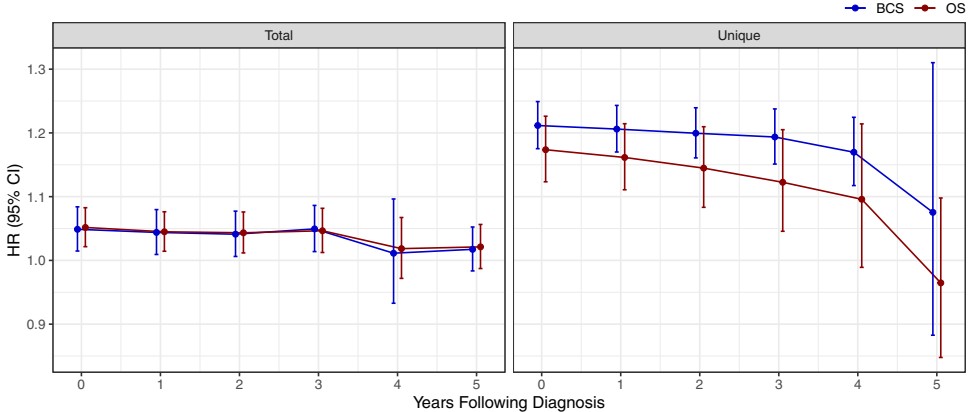

**Fig. 3 | Landmarking analysis to evaluate the impact of antimicrobial exposure on survival over time on *N* = 772 independent patients.** Data were presented as the hazard ratios ± 95% confidence intervals that reflect the risk of ongoing antimicrobial exposure at yearly intervals post-diagnosis and are plotted for the cumulative exposure definitions of total and unique antimicrobial exposures, for both overall and breast cancer-specific survival. HR hazard ratio, CI confidence interval, OS overall survival, BCS breast cancer-specific survival.

to be exposed to antimicrobials. Similarly, while sicker patients may receive more antimicrobials and have inferior survival unrelated to an antimicrobial effect, we found a comparable BCS association after controlling for severe illness that continued for three years post-diagnosis, suggesting a sustained, illness-independent effect. As all patients had non-metastatic disease at diagnosis and were treated with curative intent, the longitudinal nature of this survival association suggests ongoing antimicrobial exposure after completion of treatment impacts the risk of recurrence related to the host immune response to residual disease, which parallels the clinical timeframe of greatest risk in these women. Given that we could not profile gut taxonomy or T-cell receptor phenotypes, we could not evaluate the relationship between antimicrobial exposure and gut microbiome composition or TIL subset evolution during chemotherapy, nor the relationship between circulating and tumor-associated lymphocytes through treatment. Consistent with our previous report[44], we observed again here that unilateral mastectomy was associated with inferior survival compared to bilateral mastectomy or lumpectomy with radiation; this likely reflects unmeasured confounders, given equivalent survival after mastectomy versus breast-conserving therapy in randomized trials[45,46], and not an antimicrobial-mediated survival effect. A small portion of patients did not have a documented surgical procedure in our analyzed dataset; as surgery is the standard of care for all early-stage TNBCs, it is likely that these patients had surgeries performed outside of California, which would not be captured by the CCR, or that they had comorbidities or other factors precluding surgery. The BMI of this study sample is lower, and neighborhood socioeconomic status is higher than national averages, which reflects a local demographic and may limit the findings' generalizability.

This study's limitations are balanced by its notable strengths, including innovative study design with a large sample of TNBC patients treated in different healthcare settings; integration of EMR and CCR data which enables a rich set of covariates; near-complete mortality capture through CCR protocols; and comprehensive statistical modeling of the impact of both ALC and antimicrobial exposure on survival outcomes over time. Furthermore, these robust, clinically relevant findings support the design of future prospective studies that will collect corollary microbiome and TIL data.

This study's results are hypothesis-generating, with potential implications for patient care: they suggest careful consideration of the number and range of antimicrobials prescribed to patients with curable TNBC and other cancers. Understanding the mechanism of the observed association between antimicrobial exposure and inferior survival should be a research priority, with attention to the microbiome, TILs, and peripheral lymphocytes.

## Methods
### Dataset and patients
All research reported here, including a waiver of individual consent for research use of de-identified EMR and CCR data, was approved by institutional review boards of Stanford University, Sutter Health, and the State of California. Participants were not compensated. We used Oncoshare, a breast cancer research database that integrates electronic medical record (EMR) with California Cancer Registry (CCR) data for patients treated for breast cancer at Stanford University Healthcare and/or several sites of the community-based Sutter Health network[35,36]. EMR data were extracted, mapped to a standardized lexicon[47], and linked at the patient level to CCR records, after which all identifying information was removed. CCR mandates statewide reporting and has excellent long-term follow-up data[48]. More detailed methods of the Oncoshare project have been previously described in ref. 36.

Women were eligible for inclusion if diagnosed with stage I-III TNBC from January 2000 to May 2014; the 2014 cut-off was chosen to allow a minimum 5-year observation period for the survival endpoint. Exclusion criteria were a history of the human immunodeficiency virus (HIV) or immunocompromised status, defined as having received immunosuppressive therapy prior to breast cancer diagnosis; having unknown race or ethnicity, stage 0 disease, and no ALC measurements after diagnosis. TNBC was defined as breast cancer lacking estrogen receptor (ER) and progesterone receptor (PR) expression by immunohistochemistry (IHC) and lacking human epidermal growth factor receptor 2 (HER2) overexpression or amplification by IHC or fluorescent in situ hybridization.

### Outcomes and exposures
Clinical and sociodemographic variables derived from the CCR included: age, race and ethnicity, cancer stage, tumor grade, ER, PR and HER2 status, surgical procedure, receipt of neoadjuvant chemotherapy or radiotherapy, ever publicly insured status during the observation period, place of care, follow-up time, and neighborhood socioeconomic status (nSES). nSES was defined by Census data using the Yost index, a validated composite measure derived from block group level socioeconomic variables[49]. Variables derived from the EMR included: body mass index (BMI), ALC, ANC, use of growth factor support, antibiotic administration route, and diagnoses of systemic inflammatory response syndrome (SIRS), sepsis, or severe sepsis. For patients who did not have BMI data in the EMR, BMI was derived from

height and weight data in the CCR. Only ALC and ANC measurements recorded after breast cancer diagnosis were included in analyses and evaluated in month-intervals from diagnosis. The lowest monthly value was used for multiple measurements in the same month. Low ALC was defined as having an ALC <1 K/µL and low ANC as having an ANC <1 K/µL at any time during observation. Prescription use and laboratory measurements were assumed to be constant within month-periods where not explicitly observed. Minimum ALC and ANC post-diagnosis were defined as the minimum value observed after TNBC diagnosis.

OS and BCS were derived from a combination of the CCR, EMR, and the Stanford Cancer Institute Research Database. These were defined as months from breast cancer diagnosis until all-cause death or death due to breast cancer, respectively. Patients who did not die were censored at the date of their last follow-up. Germline *BRCA1* and *BRCA2* (*BRCA1/2*) pathogenic variant status of clinically tested patients was obtained through Myriad Genetics, Inc. (Salt Lake City, Utah).

**Antimicrobial exposure.** Only antimicrobials prescribed after breast cancer diagnosis were included in analyses and were evaluated in month-intervals from diagnosis. We excluded IV vancomycin, given its lack of gut penetration, as well as antimicrobials administered by ophthalmic, inhalation/intranasal, and intravaginal routes, given minimal systemic effect. We excluded fosfomycin, given its additional chemotherapeutic use. A full list of antimicrobials and growth factors is included in the supplement (Supplemental Table 1).

Given our interest in both breadth and intensity of antimicrobial use as potential mediators of the gut microbiome and survival, we considered three time-varying antimicrobial exposure definitions: (1) any exposure; (2) a cumulative number of total prescriptions; and (3) cumulative number of unique prescriptions, each at time $t$. For (1), the value for any exposure initiates as 0 (unexposed) for all patients not receiving an antimicrobial at time $t$ and updates to and remains as 1 for the observation period (exposed) once a prescription is observed. The count of total and unique prescriptions also initiated at 0 at the time of diagnosis, then cumulatively increased over time as further prescriptions were observed. Multiple prescriptions for the same antimicrobial were considered separate only if prescribed at least 30 days apart to account for medication refills for a single infectious episode.

**Statistical analyses**
Characteristics of patients were described overall and by exposure status (ever versus never-antimicrobial usage) using counts, percentages, medians, and interquartile ranges (IQR) and compared using a Pearson's Chi-squared test, Fisher's exact test, or Wilcoxon rank-sum test where appropriate. Time to OS or BCS and antimicrobial exposure were compared within each of the three prescription exposure definitions using Kaplan–Meier survival estimates. Continuous exposures were categorized into quartiles for survival curve visualization. Missing data were assumed to occur at random due to missing clinical visits or limitations of data recording in the EMR.

**Model of the pathway between antimicrobial use and survival.** We derived the following model to describe the relationship between antimicrobial use, ALC, and survival outcomes: ALC confounds the observed relationship between antimicrobial use and mortality because patients with lower minimum time-varying ALC may be more likely to receive antimicrobials at a given point in time. Patients with lower ALC are also more likely to have worse survival outcomes[19]. ALC may be impacted by both prior and current antimicrobial exposure and reflect antimicrobial-mediated gut microbiome dysbiosis. Conventional statistical adjustment treating ALC as a time-dependent variable in a regression model may bias the estimated effect of antimicrobial exposure toward the null, because the indirect effect of antimicrobial exposure through ALC is conditioned away by the effect of ALC on mortality. An MSM, by contrast, considers the known effect

of time-varying ALC on survival, and was fit for each exposure definition to study the pathway of interest between antimicrobial exposure and survival outcomes.

**Estimating the association between antimicrobial exposure and survival outcomes.** To estimate the association between antimicrobial use and survival outcomes in the presence of time-varying ALC, we fit an MSM to each exposure[50]. We selected variables that have known associations with OS or that were potential mediators of gut microbiome dysbiosis, which may act as confounders in analyses. These variables included: ALC[19]; age[51]; race and ethnicity[52]; socioeconomic status[49]; cancer stage, tumor grade, and receipt of chemotherapy[53]; receipt of radiotherapy[46]; use of growth factor support[54]; surgery type[44–46]; antibiotic administration route[55]; and diagnoses of SIRS, sepsis, or severe sepsis[56].

We used inverse probability weighting (IPW) to generate a "pseudo-population" representative of a hypothetical population in which antimicrobial use is allocated independent of confounding variables[57]. For each observed antimicrobial exposure, a set of stabilized weights was assigned inversely to the predicted probabilities of the corresponding exposure given baseline characteristics and longitudinally recorded variables, with the greatest weight assigned to observations with exposure and confounder combinations least represented in the sample relative to what would have been observed under random exposure.

To estimate the denominator of the weights for the MSMs, a Cox proportional hazards model of time to any antimicrobial use was used, incorporating time-varying ALC and the time-fixed baseline variables as previously described[19]. The model used to estimate the numerator of the weights was similar but excluded time-varying ALC. For the cumulative exposure definitions (total and unique antimicrobials), a similar approach was used, using Poisson regression models. Extreme weights for the continuous exposure models were trimmed to discard subjects with the extremes of small or large weights for covariate patterns potentially violating positivity[58]. To evaluate the validity of the positivity assumption in the MSM, we evaluated the mean stabilized weight and standard deviation, minimum, and maximum stabilized weights for each exposure definition.

Standard errors were based on robust variance estimates to account for the correlation of observations within a participant over time[59]. Hazard ratios (HRs) and 95% confidence intervals (CIs) were reported for time-to-event model results. We reported results from the IPW models for each exposure definition as HRs for any antimicrobial use and as the change in the number of per-month prescriptions for the continuous exposure definitions.

All analyses were conducted using R v4.0.2[60]. A $p$ value <0.05 was considered statistically significant.

**Sensitivity analysis.** To control for disease severity, we considered an acute illness, defined by receipt of intravenous antimicrobials and ICD10 codes associated with acute illness (sepsis, severe sepsis, SIRS) 90 days prior to or following antimicrobial prescriptions, as an additional covariate in the MSM model.

To validate that the observed antimicrobial-mortality effect was associated with ALC and that ALC was not a proxy for another measurement, we re-ran the MSMs that were fit as described above, replacing ALC with ANC. The results were compared to similarly adjusted unweighted Cox regression models.

**Estimating the longitudinal impact of antimicrobial exposure on survival.** To evaluate the impact of antimicrobials on mortality outcomes over time-related to time-varying ALC measurements, we performed landmark analysis based on the subset of individuals who were alive and disease-free at each landmark time point[61]. We re-computed the time-varying inverse probability weights at each landmark yearly interval following diagnosis for the cumulative antimicrobial exposure

definitions and used MSMs to determine the time-varying hazard ratios for OS and BCS.

**Exploratory analysis of baseline TIL levels, antimicrobial exposure, and response to chemotherapy.** To evaluate the association between TIL density, antimicrobial exposure, and tumor response to chemotherapy, we performed an exploratory subset analysis on 53 patients who were treated with neoadjuvant chemotherapy on the prECOG0105 study with carboplatin, gemcitabine, and iniparib[37] who had pretreatment stromal TILs scored in deciles and had antimicrobial exposure documentation in Oncoshare. Stromal TILs were considered those within the stromal portion of the slide and not in direct contact with the tumor. We evaluated the association between baseline TIL score and post-diagnosis antimicrobial exposure in the year following diagnosis, and the association between antimicrobial exposure and final pathology in a subset of 28 of these patients.

### Ethics and inclusion statement
The research includes local researchers and is locally relevant as the Oncoshare database contains data from Northern California patients. The study was approved by the institutional review boards of Stanford University, Sutter Health, and the State of California.

## Data availability
The datasets generated and/or analyzed in this study are derived from electronic medical records linked to patient-level data from the California Cancer Registry (CCR), a government-mandated program that contributes data to the larger United States Surveillance, Epidemiology and End Results (SEER) registry. Due to potential identifiability and institutional and state data-sharing policies, the full data are available upon request to the principal investigator and corresponding author, Allison Kurian, with a data use agreement facilitated through the Stanford University Office of Research Administration (https://ora.stanford.edu/).

## Code availability
Custom code in R generated for analysis of the data is available at: https://github.com/vsritter/TNBC_MSM.

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

## Acknowledgements

This work was supported by the Breast Cancer Research Foundation, the Susan and Richard Levy Gift Fund, the Suzanne Pride Bryan Fund for Breast Cancer Research, the Jan Weimer Junior Faculty Chair in Breast Oncology, the Regents of the University of California's California Breast Cancer Research Program (16OB-0149 and 19IB-0124), the BRCA Foundation, the G. Willard Miller Foundation, and the Biostatistics Shared Resource of the NIH-funded Stanford Cancer Institute (P30CA124435). The collection of cancer incidence data used in this study was supported by the California Department of Public Health pursuant to California Health and Safety Code Section 103885; the Centers for Disease Control and Prevention's National Program of Cancer Registries, under Cooperative Agreement No. 5NU58DP006344; and the National Cancer Institute's SEER Program under Contract No. HHSN261201800032I awarded to the University of California, San Francisco, Contract No. HHSN261201800015I awarded to the University of Southern California, and Contract No. HHSN261201800009I awarded to the Public Health Institute, Cancer Registry of Greater California. K.A. was supported by NIH 5T32HG000044. This work was further supported by a Stand Up 2 Cancer grant, a V Foundation Fellowship, and Damon Runyon Clinical Investigator Award and NIH R01AI143757 (to A.S.B.). The ideas and opinions expressed herein are those of the authors and do not necessarily reflect the opinions of the State of California, the Department of Public Health, the National Cancer Institute, the Centers for Disease Control and Prevention, or their contractors and subcontractors.

## Author contributions

J.D.R., K.A., A.S.B., and A.W.K. conceived and designed the study. A.W.K., M.L., S-Y.L., and S.L.G. contributed to the collection and assembly of data. J.D.R., V.R., N.P., S.H., A.W.K., and A.S.B. analyzed and interpreted the data. J.D.R., V.R., S.H., and N.P. performed statistical analyses. J.D.R., A.W.K., and A.S.B. drafted the manuscript. S.H., S-Y.L., E.M.J., S.L.G., M.L.T., L.S., H.I., and G.W.S. provided critical revisions to the manuscript.

## Competing interests

The authors declare no competing interests.
