## [Peer Review File · Nature Communications]

REVIEWER COMMENTS

Reviewer #2 (Remarks to the Author):

Thanks very much for addressing all my suggestions and comments

Reviewer #3 (Remarks to the Author):

The authors kindly addressed all raised comments and in fact, the major strength, which is novelty of findings, has been pointed out. However, I consider the lack of TIL data reflecting a well accepted metric of tumor specific immunity a major limitation, which is not sufficiently addressed by adding data for 53 out of 772 patients analyzed. To meet exceptional scientific quality criteria, one may indeed expect to analyze all tissues. It is likely that almost all FFPE tissue blocks are archived and can be scored for TILs by most if not all pathologists according to guidelines of the international TIL working group.

Reviewer #4 (Remarks to the Author):

MAJOR COMMENTS

1. Overall: The authors do not comment on the assumptions of their use of propensity scores. While some acknowledgement of the assumptions of consistency and exchangeability should be in the Discussion section, the Methods and Results should include evaluation of the positivity assumption and ensure there is no misspecification of the model used to estimate the weights.
2. When examining Supplemental Figure 2, it is clear that number of total antimicrobial prescriptions and Number of unique antimicrobial prescriptions are discrete count distributions that are highly skewed. However, the authors stage on page 5 in the Probabilities of antimicrobial use section the

authors state that mixed effect[s] linear regression models were used and page 14 line 430 states that linear regression models were used. It seems unlikely that model assumptions would be met and there is no discussion of how assumptions were assessed. An appropriate discrete model should be used instead. The marginal structural model for cumulative and unique exposures should be refit accordingly.

3. Supplemental Tables 3, 4, 5, 6, Figures 1, 2 and Results will need to be redone after fitting an appropriate model as a linear model likely does not meet assumptions for total and unique antimicrobial exposures. It is also unclear why in Supplemental Table 6 the unique antimicrobial exposures N is so large – the sample size is unique patients not exposures.

4. Page 6, Associations of antimicrobial exposure with survival over time; lines 455-458: It is not clear why the authors changed to a logistic regression model when they have a time to event outcome. It seems more reasonable to include an interaction with time and report the HRs at the timepoints of interest rather than fit a logistic regression model and report odds ratios. To support the statement in the discussion on lines 217-218 “decreased in strength over time” a plot of HR over time could be presented.

MINOR COMMENTS

5. Page 5, Probabilities of antimicrobial use: This section title is confusing given the models fit. Also, the last sentence, “results were similar for total and cumulative unique antimicrobial use” should be further clarified as one may infer the similarity is with the HRs in time to first use versus here the authors are modeling number, so the results are similar to one another.

6. Sensitivity analysis, page 6: It would be helpful if the unweighted and weighted or other adjusted models appeared together in one table side-by-side to facilitate comparisons. Also, page 14 lines 444-447 states they replaced ALC with ANC. It would be more useful to check the HR if ALC were included and not included in the model.

7. Page 7, Exploratory analysis: Given the limited sample size, the Discussion on lines 237-240 should be tempered because the authors were not necessarily powered to detect differences.

8. Page 7, Discussion: On line 214 the authors state there were “multiple institutions representing community and academic practice.” But Page 4 states patients were treated at either Stanford University of Palo Alto Medical Foundation (Sutter Health). Please clarify.

We thank the initial reviewers of this manuscript for their feedback on our revisions, and Reviewer #4 for the thorough review of its statistical methods. We have responded to all the Reviewers' comments point-by-point in the response to reviewers comments below, and additionally thank the *Nature Communications* Editorial Office for this invitation to revise and improve the manuscript further.

Reviewer #2 (Remarks to the Author):

Thanks very much for addressing all my suggestions and comments

We thank Reviewer #2 for their initial suggestions for improving our work and are pleased that we were able to address the suggestions and comments in a satisfactory manner.

Reviewer #3 (Remarks to the Author):

The authors kindly addressed all raised comments and in fact, the major strength, which is novelty of findings, has been pointed out. However, i consider the lack of TIL data reflecting a well-accepted metric of tumor specific immunity a major limitation, which is not sufficiently addressed by adding data for 53 out of 772 patients analyzed. To meet exceptional scientific quality criteria, one may indeed expect to analyze all tissues. It is likely that almost all FFPE tissue blocks are archived and can be scored for TILs by most if not all pathologists according to guidelines of the international TIL working group.

We appreciate the remarks of Reviewer #3, noting that the novel findings of our work are clearer in the revised manuscript. We do agree that TILs analysis of a larger portion of this study sample would be of interest, as TILs are a validated measure of tumor-directed immunity; however, we cannot obtain archived post-neoadjuvant blocks for the majority of this cohort for several reasons:

- (1) The standard-of-care treatment for early-stage triple-negative breast cancer has evolved toward greater use of neoadjuvant (pre-surgery) chemotherapy over the past decade, but only a minority of patients in this study sample were treated with neoadjuvant chemotherapy (26%, N = 200). Therefore, for most patients treated in the early 2000s with adjuvant (post-surgery) chemotherapy, we do not have tissue samples from after receipt of chemotherapy and antimicrobials in which to analyze TILs.
- (2) Even among the subset treated with neoadjuvant chemotherapy, pathology blocks are not typically kept beyond 5 years if not part of a clinical trial (as were the 53 analyzed patients of these 200), and so the majority of these cases do not have recoverable tissue blocks, due to when surgeries were performed.
- (3) Approximately half of patients were treated outside of Stanford, where pathology blocks are not available for us to request.

We echo the sentiment of Reviewer #3 that prospectively collected, post-neoadjuvant TILs analysis – that is, TIL changes after antimicrobial exposure – is of great interest, with additional attention to how immune checkpoint inhibitor exposure alters TIL dynamics through chemotherapy. Given the above, we anticipate this type of analysis would best occur through a clinical trial, through which corollary fecal samples for microbiome analysis could also be collected. In such a setting, we could analyze not only quantitative TIL density changes, but also shifts in T cell receptor repertoires though treatment and with antimicrobial exposure. Such analyses would provide important tissue correlate data to support the mortality associations highlighted in this work. We anticipate this manuscript will serve as motivation for such a study.

Reviewer #4 (Remarks to the Author):

MAJOR COMMENTS

1. Overall: The authors do not comment on the assumptions of their use of propensity scores. While some acknowledgement of the assumptions of consistency and exchangeability should be in the Discussion section, the Methods and Results should include evaluation of the positivity assumption and ensure there is no misspecification of the model used to estimate the weights.

We thank Reviewer #4 for their thorough review of the statistical rigor of our manuscript and for important points raised here regarding assumptions made for the use of propensity scores. We have revised the Methods and Results to include an evaluation of the positivity assumption to ensure no model misspecification, and the Discussion section to acknowledge assumptions of consistency and exchangeability.

To briefly summarize, we conducted the following additional analyses to evaluate the positivity assumption for the propensity scores (i.e., inverse probability weights for the three exposure definitions in our study) based on prior work (Austin and Stuart 2015): we assessed the mean stabilized weight and standard deviation of the stabilized weights for each exposure definition, along with the minimum and maximum. We evaluated the proximity of the mean value of the stabilized weight to one, as values deviating from one could indicate non-positivity or model misspecification. We trimmed the extremes of small or large weights to discard subjects with covariate patterns potentially violating positivity (as per Austin and Stuart 2015). In the revised manuscript, we report the following statistics for the stabilized weights for the three exposure definitions that meet the positivity assumption:

1. For inverse probability weighting for any antimicrobial exposure definition, the mean stabilized weight was 1.0, and the standard deviation of the stabilized weights was 0.08. The minimum and maximum weights were 0.7 and 1.5, respectively.
2. For total antimicrobial exposure, the mean stabilized weight was 1.1 and the standard deviation of the stabilized weights was 0.6. The minimum and maximum weights were 0.4 and 3.2, respectively.
3. For unique antimicrobial exposure, the mean stabilized weight was 1.1 and the standard deviation of the stabilized weights was 0.3. The minimum and maximum weights were 0.6 and 3.2, respectively.

We have updated the Methods and Results sections detailing the above, as well as the Discussion section, to address the assumptions of consistency and exchangeability in the section describing study limitations.

2. When examining Supplemental Figure 2, it is clear that number of total antimicrobial prescriptions and Number of unique antimicrobial prescriptions are discrete count distributions that are highly skewed. However, the authors state on page 5 in the Probabilities of antimicrobial use section the authors state that mixed effect[s] linear regression models were used and page 14 line 430 states that linear regression models were used. It seems unlikely that model assumptions would be met and there is no discussion of how assumptions were assessed. An appropriate discrete model should be used instead. The marginal structural model for cumulative and unique exposures should be refit accordingly.

We appreciate Reviewer #4's thorough review and feedback. To address the Reviewer's concern, we have refit the inverse probability weights for total and unique antimicrobial exposures incorporating the Poisson distributions (rather than the Gaussian models) and have updated the results for MSMs analyses, tables, and figures throughout the manuscript. Importantly, we note that the analyses of the stabilized weights for evaluating the positivity assumption in the response to the previous comment (Major Comment #1) are based on these new weights incorporating the Poisson models. The updated results are nearly unchanged, with minimal changes to the reported hazard ratios and this study's main findings.

3. Supplemental Tables 3, 4, 5, 6, Figures 1, 2 and Results will need to be redone after fitting an appropriate model as a linear model likely does not meet assumptions for total and unique antimicrobial exposures. It is also unclear why in Supplemental Table 6 the unique antimicrobial exposures N is so large – the sample size is unique patients not exposures.

As above, we have updated Figures 1 and 2, Supplemental Tables 3, 4, 5, 6 (and new Supplemental Table 7), and the manuscript text to include results of the updated inverse probability weighting and MSMs using the Poisson models. The number of unique antimicrobial exposures in revised Supplemental Table 6 was incorrectly reported in the initial table; it has been updated to N = 197 events. We appreciate Reviewer #4 noting this error.

4. Page 6, Associations of antimicrobial exposure with survival over time; lines 455-458: It is not clear why the authors changed to a logistic regression model when they have a time to event outcome. It seems more reasonable to include an interaction with time and report the HRs at the timepoints of interest rather than fit a logistic regression model and report odds ratios. To support the statement in the discussion on lines 217-218 “decreased in strength over time” a plot of HR over time could be presented.

We appreciate Reviewer #4's suggestion. We initially used a pooled logistic regression model that can approximate hazard ratios of time-dependent Cox models (D'Agostino, Lee et al. 1990), based on our prior work on the impact of ALC and mortality (Afghahi et al. 2018). However, given that the Cox models used in our study meet the proportional hazards assumption (though we note detecting a departure from such an assumption can be underpowered (Stensrud and Hernán 2020)) and given that Reviewer #4 raised concerns related to this pooled logistic regression approach for evaluating time-dependent effects, we have conducted an updated analysis based on Cox regression by applying a landmarking approach (Putter and van Houwelingen 2017). This landmarking analysis helps answer the question of whether the effect of antimicrobial exposure measured k years post diagnosis ($k = 1, 2, 3, 4, 5$ years)—in subjects alive at each landmark timepoint—differs from the effect measured at diagnosis. In contrast to time-dependent Cox regression analysis, this landmarking analysis does *not* answer whether exposure measured at the time of diagnosis (or later if time-varying covariates are considered) has a time-dependent effect at later time points. Instead, it considers the effect of *ongoing* exposure on the subset of women who remain event-free at k years after diagnosis. This updated analysis better answers the question of how exposure over time impacts survival; it is also clinically meaningful since it addresses whether exposure measured at later time points—after completion of chemotherapy with curative intent—impacts the ongoing risk of recurrence and the impact of antimicrobials on the immune response to residual disease.

We have generated a new Figure 3 to replace the prior Supplemental Table 7, plotting the hazard ratios over time against each landmarking point (i.e., 0, 1, 2, 3, 4, 5 years after diagnosis) for the cumulative antimicrobial exposure definitions (total and unique exposures) that are associated

with decreased survival through year 5 of observation for both overall and breast cancer-specific survival. Here, we see a sustained association of exposure with survival through year 3 post-diagnosis that then decreases at years 4 and 5, suggesting ongoing exposure impacts the risk of cancer recurrence related to the host immune response to residual disease that persists after completion of treatment with curative intent.

MINOR COMMENTS

5. Page 5, Probabilities of antimicrobial use: This section title is confusing given the models fit. Also, the last sentence, “results were similar for total and cumulative unique antimicrobial use” should be further clarified as one may infer the similarity is with the HRs in time to first use versus here the authors are modeling number, so the results are similar to one another.

We have revised the title of this section to “Inverse probability weighting to estimate probabilities of antimicrobial use” to better reflect the models fit. For conciseness and improved clarity, we have revised the text in this section to highlight which covariates were associated with antimicrobial use for each exposure definition, and have referred readers to the supplementary tables for the hazard ratios and estimates.

6. Sensitivity analysis, page 6: It would be helpful if the unweighted and weighted or other adjusted models appeared together in one table side-by-side to facilitate comparisons. Also, page 14 lines 444-447 states they replaced ALC with ANC. It would be more useful to check the HR if ALC were included and not included in the model.

Based on Reviewer #4’s suggestion, we have generated a new Supplemental Table 7 comparing the unweighted Cox regression model to the MSM, considering the impact of both absolute lymphocyte count and of absolute neutrophil count separately. This table permits comparison of how the HRs change when ALC or ANC is not included (in the unweighted Cox model) versus is included (in the MSM) in the model. This updated table highlights how the HRs for OS and BCS change when considering the known impact of ALC on mortality (as shown in our prior work: Afghahi et al. 2018) in the MSM, but do not change when considering the impact of ANC, suggesting that ALC but not ANC is associated with both overall survival and breast cancer-specific survival in the MSM model used throughout the manuscript.

7. Page 7, Exploratory analysis: Given the limited sample size, the Discussion on lines 237-240 should be tempered because the authors were not necessarily powered to detect differences.

We agree that the analysis was not sufficiently powered to detect an association between antimicrobial exposure and surgical pathology, and therefore that we cannot draw a conclusion about this association without a larger dataset (unavailable for this patient cohort, as detailed above in response to comments from Reviewer #3). We have revised the text in the discussion section highlighted by Reviewer #4 to indicate that the analyzed sample set was small in size and therefore that analysis was limited in power.

8. Page 7, Discussion: On line 214 the authors state there were “multiple institutions representing community and academic practice.” But Page 4 states patients were treated at either Stanford University of Palo Alto Medical Foundation (Sutter Health). Please clarify.

We have revised the text to clarify that our study sample consisted of patients from two institutions representing community (Sutter Health) and academic (Stanford University) practice.

References

Afghahi, A., et al (2018) “Higher Absolute Lymphocyte Counts Predict Lower Mortality from Early-Stage Triple-Negative Breast Cancer.” Clin Cancer Res **24**: 2851-2858.

Austin, P. C. and E. A. Stuart (2015). "Moving towards best practice when using inverse probability of treatment weighting (IPTW) using the propensity score to estimate causal treatment effects in observational studies." Statistics in medicine **34**(28): 3661-3679.

D'Agostino, R. B., et al. (1990). "Relation of pooled logistic regression to time dependent Cox regression analysis: the Framingham Heart Study." Statistics in medicine **9**(12): 1501-1515.

Putter, H. and H. C. van Houwelingen (2017). "Understanding landmarking and its relation with time-dependent Cox regression." Statistics in biosciences **9**: 489-503.

Stensrud, M. J. and M. A. Hernán (2020). "Why test for proportional hazards?" Jama **323**(14): 1401-1402.

REVIEWERS' COMMENTS

Reviewer #4 (Remarks to the Author):

The authors have addressed my previous comments and I have no further concerns.

We thank Reviewer #4 for their thoughtful remarks on our manuscript, "Antimicrobial exposure is associated with decreased survival in triple-negative breast cancer," and the Nature Communications Editorial Office for the opportunity to submit our revised manuscript.

Reviewers' Comments:

Reviewer #4 (Remarks to the Author):

The authors have addressed my previous comments and I have no further concerns.

We thank Reviewer #4 for their thorough review of the statistical rigor of our manuscript and excellent suggestions for its improvement. We are pleased that our revisions were well-received.